# Non-linear association between neutrophil-to-lymphocyte ratio and 90-day mortality in patients with pneumonia receiving glucocorticoids alone or in combination with other immunosuppressants: A retrospective cohort study

Caizhen Chen *, Xiuguo Zhang, Baoli Li, Qian Geng

Department of Nursing, Third Hospital of Hebei Medical University, Shijiazhuang, Hebei, China

* amily_ccz@163.com

## Abstract

### Objective

The relationship between neutrophil-to-lymphocyte ratio (NLR) and 90-day mortality in patients with pneumonia receiving glucocorticoids alone or in combination with other immunosuppressants has not been fully verified. We aimed to explore the influence of NLR on 90-day mortality in this specific population.

### Methods

This study utilized the data set from the Dryad database, involving 696 participants diagnosed with pneumonia who were receiving glucocorticoids alone or in combination with other immunosuppressants. Data on demographics, vital signs, laboratory results, and comorbidities were collected to assess the link between NLR and 90-day mortality. Multivariable Cox hazard regression analyses and smooth curve fitting were employed to assess the independent association between NLR and 90-day mortality. A two-piecewise linear regression model was used to examine the nonlinear association between NLR and in-hospital mortality. Receiver-operating characteristic curves (ROC) and area under the curves (AUC) were used to assess the ability of different biomarkers to predict the 90-day mortality in patients with pneumonia.

### Results

In total, 696 patients with pneumonia were included in this study. There were 332 individuals (47.7%) aged 18–59 years and 364 (52.3%) aged 60–99 years; 52.6% were male. The 90-day mortality rate across the study population was 26.1%. A non-linear association was noted between NLR and 90-day mortality, with an

**Data availability statement:** The dataset can be obtained from the Dryad Digital Repository (https://doi.org/10.5061/dryad.mkkwh70x2).

**Funding:** The author(s) received no specific funding for this work.

**Competing interests:** The authors have declared that no competing interests exist.

inflection point at approximately 16.475. On the left side of the inflection point, the hazard ratio was 1.145(95% confidence interval [CI]: 1.091–1.2, $p < 0.001$). On the right side of the inflection point, the hazard ratio was 1.0057(95% CI:0.9923–1.0192; $p = 0.406$), reflecting a lack of statistical significance. Similar patterns were observed in subgroup analyses, with significant interaction effects noted for age and smoking status. Furthermore, the ROC curve analysis revealed that NLR was the optimal biomarker for predicting the 90-day mortality with an AUC of 0.714 (95% CI:0.670–0.757). Using 9.34 as the cutoff value of NLR, the sensitivity was 69.8%, and the specificity was 67.7%.

## Conclusions

A nonlinear correlation between NLR and 90-day mortality was identified in pneumonia patients undergoing glucocorticoid treatment. The NLR value of 16.475 represented the optimal threshold for predicting the 90-day mortality, after exceeding the threshold,90-day mortality tended to stabilize. The findings suggest that NLR is a practical and useful biomarker for predicting the 90-day mortality in this population.

## Introduction

Extended administration of high-dose glucocorticoids is associated with significant immunosuppression and increased susceptibility to severe infections, particularly pulmonary infections, which are a predominant cause of morbidity and mortality in this immunocompromised population [1]. The mortality rate from pulmonary infections in patients receiving prolonged glucocorticoid therapy approximates 45% [1–4], with strikingly similar mortality patterns observed in patients presenting with other etiologies of immunosuppression [5]. Globally, lower respiratory tract infections represent a substantial public health concern, with epidemiological data from 2019 documenting 489 million reported cases [6]. In the United States, pneumonia remains the ninth leading cause of mortality and the most prevalent infectious disease-related cause of death, resulting in an estimated 50,000 annual fatalities [7].

Given the high incidence and mortality of pulmonary infections in immunocompromised patients, identifying reliable biomarkers to predict the severity and outcomes of these infections is crucial. Because neutrophils and lymphocytes represent the largest percentage of all immune cells circulating in the bloodstream, the neutrophil-to-lymphocyte ratio (NLR), a hematological parameter that reflects systemic immune status and inflammatory responses, has recently emerged as a promising biomarker owing to its cost-effectiveness, rapid assessment, and widespread clinical availability, particularly in the context of chronic disease management [8–11]. NLR is calculated as the ratio of neutrophil count to lymphocyte count, typically from a peripheral blood sample. Under pathological stress conditions, neutrophil counts demonstrate a marked elevation, whereas lymphocyte populations show a substantial decline. Elevated NLR values have been observed in various infectious diseases, including acute

exacerbations of chronic obstructive pulmonary disease (COPD) [12], pneumonia [13], coronavirus disease 2019 (COVID-19) [14,15], and sepsis [16], and have been associated with more severe disease and increased mortality risk [17,18].

Despite these findings, the association between NLR and mortality risk in patients with pneumonia receiving glucocorticoids remains unclear. Therefore, this study aimed to investigate the potential correlation between the NLR and 90-day mortality within this specific patient cohort and hypothesized that a higher NLR is associated with increased mortality in patients with pneumonia.

## Materials and Methods

### Study population

Dryad is an open data knowledge base that stores medical, biological, and ecological data. The data used in this study were derived from an observational study conducted in six secondary and tertiary academic hospitals in China [19], supplemented by a dataset obtained from the Dryad Digital Repository (https://doi.org/10.5061/dryad.mkkwh70×2). The Institutional Review Board of the China-Japan Friendship Hospital (Approval No.2015−86) authorized this retrospective investigation and oversaw centralized coordination among the participating institutions while ensuring standardized protocols for anonymized data collection and submission. Informed consent was waived due to the retrospective nature of the study.

The study enrolled 696 consecutive patients diagnosed with pneumonia who were hospitalized across six secondary and tertiary academic medical centers in China between January 1, 2013, and December 31, 2017. This cohort study adhered to the Strengthening the Reporting of Observational Studies in Epidemiology (STROBE) guidelines, ensuring rigorous methodological standards and transparent reporting of observational research [20].

The diagnosis of pneumonia was established in accordance with the joint guidelines issued by the American Thoracic Society and Infectious Diseases Society of America [21,22]. Diagnostic criteria encompassed the presence of newly identified pulmonary infiltrates on chest radiography or computed tomography (CT), combined with at least one of the following clinical features: (1) acute onset or exacerbation of respiratory symptoms, including productive cough with purulent sputum, with or without associated chest pain; (2) documented fever (axillary temperature ≥ 37.3°C) or hypothermia (axillary temperature<36°C); (3) physical examination findings suggestive of pulmonary consolidation and/or auscultatory crackles; or (4) leukocyte count abnormalities(>$10 \times 10^9$/L or <$4 \times 10^9$/L), irrespective of neutrophil predominance. We included patients presenting with connective tissue diseases, idiopathic interstitial pneumonia, nephrotic syndrome, chronic glomerulonephritis, COPD, bronchial asthma, or those undergoing alternative immunosuppressive therapies.

Patients were selected for the study based on the following inclusion criteria: (1) treatment with oral or intravenous corticosteroids treatment [4,23,24] prior to admission;(2) confirmed pneumonia diagnosis at admission or during the hospital stay; (3) aged ≥16 years at enrollment. The exclusion criteria were as follows: (1) presence of noninfectious pulmonary pathologies, including malignant pulmonary neoplasms, noninfectious interstitial pulmonary disorders, pulmonary thromboembolism, or congestive cardiac failure; and (2) unable to provide informed consent for procedures, such as bronchoalveolar lavage(BAL).

### Data collection

A multidisciplinary team of principal investigators, comprising clinical specialists, biostatisticians, microbiologists, and radiologists, collaboratively designed the study protocol and implemented a standardized case report form (CRF) to ensure methodological consistency across all participating centers. Before the study began, all investigators received training on the protocol, including the screening process, disease definitions, and CRF use. After the data collection, a trained researcher reviewed the CRFs to ensure completeness and quality.

A comprehensive set of clinical data was systematically collected from hospitalized patients' medical records, encompassing:(1) demographic characteristics, including age and sex; (2) clinical manifestations, such as fever, cough, and

dyspnea; (3) comorbidities, including asthma, COPD, coronary heart disease (CHD), diabetes mellitus (DM), and chronic renal failure; (4) disease severity, assessed using the Pneumonia Severity Index (PSI) score; (5) laboratory findings, including platelet count, aspartate aminotransferase (AST) levels, creatinine levels, and sodium (Na) levels; and (6) survival status at 30 and 90 days after admission.

## Statistical analysis

Patients were grouped into quartiles according to the NLR distribution. The baseline characteristics were summarized using descriptive statistical methods. For continuous variables exhibiting normal distribution, the data are presented as mean ± standard deviation (SD), whereas those with skewed distributions are described using median and interquartile range (IQR). To evaluate the differences among the groups, categorical variables were analyzed using either the chi-squared test or Fisher's exact test. Normally distributed continuous variables were examined using one-way analysis of variance (ANOVA), whereas non-normally distributed continuous variables were evaluated using the Kruskal-Wallis H test.

Cox regression analyses were performed to examine the relationship between NLR and mortality within 90 days. Multiple adjusted models were developed, with the extended Cox model applied, utilizing the lowest quartile of NLR as the reference category. Model 1 represented the crude analysis without any adjustments. Model 2 incorporated adjustments for demographic variables including age and sex. Model 3 included the key clinical comorbidities of asthma, COPD, CHD, DM, and CRF. The final and most comprehensive model (Model 4) was adjusted for all aforementioned variables, in addition to lifestyle factors (smoking status and alcohol consumption), laboratory parameters(platelet count [PLT], aspartate aminotransferase [AST], creatinine [CRE], and serum sodium [Na]), and disease severity as measured by the PSI.

In addition, to investigate the dose-response association linking NLR to mortality within 90 days, smooth curve fitting was implemented. A likelihood ratio test was conducted to compare the two-piecewise linear model with the one-line linear regression model. Survival outcomes were analyzed using Kaplan–Meier survival curves, with stratification by NLR quartiles and statistical significance assessed via the log-rank test. To evaluate the robustness of the NLR-mortality link, stratified and interaction analyses were carried out across multiple subgroups, encompassing demographic factors (age, sex), comorbid conditions (asthma, COPD, CHD, DM), and behavioral factors (smoking status, alcohol consumption). Additionally, a sensitivity analysis was performed using patients with an NLR between 0.022 and 3.65 as the reference group. Patients with an NLR above 3.65 were further categorized into three groups: mild NLR (3.65–7.66), moderate NLR (7.66–15.46), and severe NLR (15.46–222.92). The Youden index was applied to determine the optimal cut-off value for each variable, and receiver operating characteristic (ROC) curve analysis assessed the sensitivity and specificity of different biomarkers predicting adverse prognosis. The area under the curve (AUC) was used to evaluate prognostic accuracy, with comparisons conducted via a nonparametric approach. Higher AUC values indicate greater discriminatory ability. To assess the robustness of our findings, sensitivity analysis was conducted by reclassifying NLR according to the established optimal cutoff value of 10.0, derived from previous studies [17,32]. We used multiple imputations, based on 5 replications and a chained equation approach method in the R MI procedure, to maximize statistical power and minimize bias that might occur to account for missing data [25]. The details of the missing value are shown in S1 Table.

All statistical analyses were conducted using R Statistical Software (version 4.2.2, http://www.R-project.org, The R Foundation) and Free Statistics Analysis Platform (version 2.1.1, Beijing, China, http://www.clinicalscientists.cn/freestatistics) [26]. A two-tailed test was used to determine statistical significance, with the threshold set at $p < 0.05$.

## Results

### Participants' characteristics

Initially, there were 716 patients with pneumonia receiving glucocorticoids in the Dryad database. After excluding missing values, 696 patients were included in the study, with 332 (47.7%) aged between 18 and 59 years, and 364 (52.3%) aged between 60 and 99 years (see Fig 1 for details). Additionally, 52.6% of the participants were male. The predominant

clinical manifestations observed in the study cohort were cough (87.8%), followed by fever (74.6%) and dyspnea (60.3%), respectively. The 90-day mortality rate for the patients was 26.1%. The baseline characteristics of all participants are shown in Table 1.

## Association between NLR and mortality

To examine the association between NLR and 90-day mortality in patients with pneumonia, we constructed a series of multivariate regression models with progressive adjustments (Table 2). Model 1 was unadjusted, whereas Model 2 was adjusted for age and sex. Model 3 was additionally adjusted for comorbidities. The fully adjusted Model 4 included all the aforementioned variables, along with smoking status, alcohol consumption, PLT, AST, CRE, Na levels, and PSI (Table 2).

As a continuous variable, in the fully adjusted multivariate regression model (Model 4), the 90-day mortality increased by 11% for every 10-unit increase in NLR(hazard ratio[HR]=1.11,95% confidence interval [CI]=1.06–1.16, $p < 0.001$).

When categorized into NLR quartile, comprehensive multivariate regression analysis (Model 4) revealed a persistent significant association with 90-day mortality. Patients in the highest quartile (Q4) exhibited a substantially elevated 90-day mortality compared with those in the lowest quartile (Q1) (Q4: HR=6.06,95% CI:3.44–10.7, $p < 0.001$). Notably, all four models demonstrated statistically significant trend associations ($p < 0.05$).

## Nonlinearity relationship between NLR and mortality

We revealed a non-linear relationship between the NLR and 90-day mortality, as determined through the application of a multivariate Cox hazard regression model and smooth curve fitting (Fig 2). Given the significant nonlinearity ($p < 0.001$)

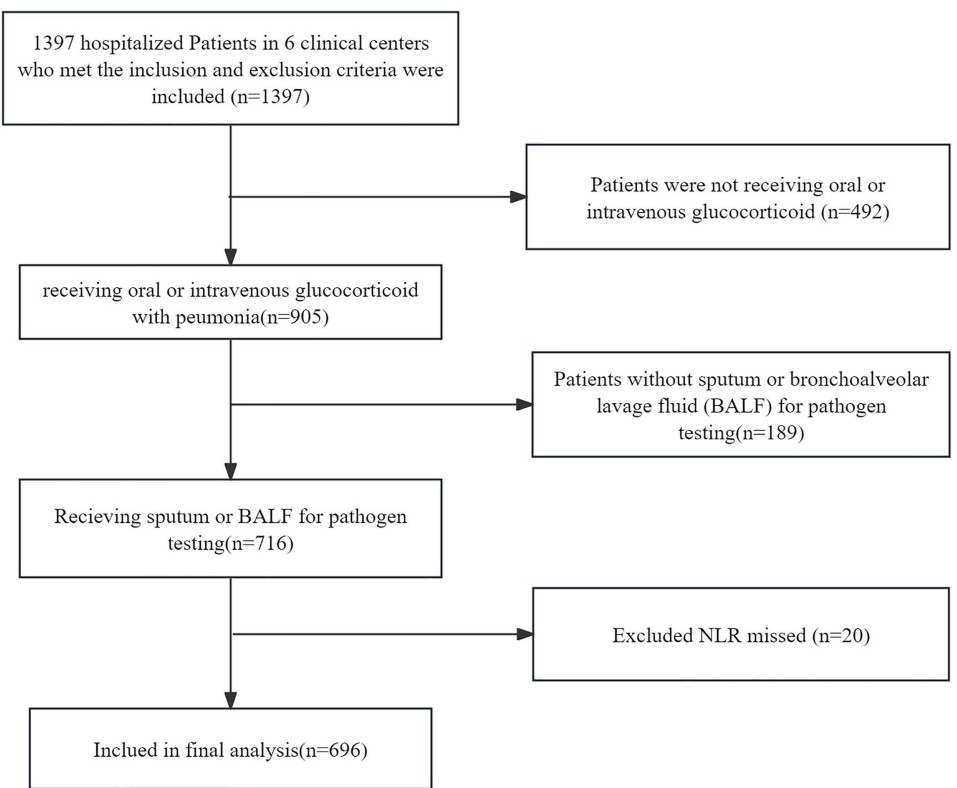

**Fig 1. Flow chart of the study.**

**Table 1. Baseline and clinical characteristics of the study population.**

| Variables | Total (n = 696) | Q1(0.02–3.64) (n = 174) | Q2(3.65–7.65) (n = 174) | Q3(7.66–15.45) (n = 174) | Q4(15.46–222.92) (n = 174) | P value |
|---|---|---|---|---|---|---|
| NLR, Median (IQR) | 0.8 (0.4, 1.5) | 0.2 (0.2, 0.3) | 0.6 (0.5, 0.7) | 1.1 (0.9, 1.3) | 2.4 (1.8, 3.3) | <0.001 |
| Age, n (%) | | | | | | 0.066 |
| 18–59 | 332 (47.7) | 73 (42) | 83 (47.7) | 97 (55.7) | 79 (45.4) | |
| 60–99 | 364 (52.3) | 101 (58) | 91 (52.3) | 77 (44.3) | 95 (54.6) | |
| Sex, n (%) | | | | | | 0.342 |
| Male | 366 (52.6) | 85 (48.9) | 86 (49.4) | 97 (55.7) | 98 (56.3) | |
| Female | 330 (47.4) | 89 (51.1) | 88 (50.6) | 77 (44.3) | 76 (43.7) | |
| Asthma, n (%) | | | | | | 0.133 |
| No | 679 (97.6) | 168 (96.6) | 167 (96) | 171 (98.3) | 173 (99.4) | |
| Yes | 17 (2.4) | 6 (3.4) | 7 (4) | 3 (1.7) | 1 (0.6) | |
| COPD, n (%) | | | | | | 0.206 |
| No | 595 (85.5) | 141 (81) | 148 (85.1) | 152 (87.4) | 154 (88.5) | |
| Yes | 101 (14.5) | 33 (19) | 26 (14.9) | 22 (12.6) | 20 (11.5) | |
| CHD, n (%) | | | | | | 0.537 |
| No | 610 (87.6) | 150 (86.2) | 149 (85.6) | 157 (90.2) | 154 (88.5) | |
| Yes | 86 (12.4) | 24 (13.8) | 25 (14.4) | 17 (9.8) | 20 (11.5) | |
| DM, n (%) | | | | | | 0.358 |
| No | 523 (75.1) | 125 (71.8) | 130 (74.7) | 139 (79.9) | 129 (74.1) | |
| Yes | 173 (24.9) | 49 (28.2) | 44 (25.3) | 35 (20.1) | 45 (25.9) | |
| CRF, n (%) | | | | | | 0.473 |
| No | 643 (92.4) | 163 (93.7) | 162 (93.1) | 156 (89.7) | 162 (93.1) | |
| Yes | 53 (7.6) | 11 (6.3) | 12 (6.9) | 18 (10.3) | 12 (6.9) | |
| Smoke, n (%) | | | | | | 0.691 |
| Never smoke | 509 (73.1) | 130 (74.7) | 120 (69) | 133 (76.4) | 126 (72.4) | |
| Former smoke | 160 (23.0) | 36 (20.7) | 46 (26.4) | 35 (20.1) | 43 (24.7) | |
| Current smoke | 27 (3.9) | 8 (4.6) | 8 (4.6) | 6 (3.4) | 5 (2.9) | |
| Alcoholism, n (%) | | | | | | 0.021 |
| No | 639 (91.8) | 163 (93.7) | 159 (91.4) | 166 (95.4) | 151 (86.8) | |
| Yes | 57 (8.2) | 11 (6.3) | 15 (8.6) | 8 (4.6) | 23 (13.2) | |
| PLT × 10⁹/L | 190.5 ± 90.7 | 179.7 ± 86.0 | 202.6 ± 90.0 | 196.1 ± 93.5 | 183.5 ± 92.1 | 0.064 |
| AST(U/L) | 24.0 (16.0, 39.0) | 20.6 (16.0, 28.8) | 22.0 (16.0, 37.0) | 25.0 (16.7, 43.5) | 27.0 (18.0, 43.8) | 0.001 |
| CRE(μmmol/L) | 64.0 (50.9, 90.3) | 63.4 (53.3, 82.6) | 63.4 (49.0, 87.6) | 63.2 (49.8, 92.0) | 68.1 (49.6, 106.6) | 0.537 |
| Na(mmol/L) | 137.6 ± 7.5 | 139.5 ± 8.5 | 138.1 ± 9.2 | 137.0 ± 5.2 | 135.8 ± 5.9 | < 0.001 |
| PSI | 81.1 ± 31.6 | 73.7 ± 27.3 | 78.1 ± 30.6 | 79.2 ± 31.1 | 93.3 ± 33.7 | < 0.001 |

Abbreviations: COPD, chronic obstructive pulmonary disease; CHD, coronary heart disease; DM, diabetes mellitus; CRF, coronary renal failure; PLT, platelet; AST, Aspartate Aminotransferase; CRE, creatinine; PSI, pneumonia severity index (PSI) score.

(Table 3), we utilized a two-piecewise linear model to characterize the relationship between NLR and 90-day mortality; a significant inflection point was identified at 16.475 (see Fig 2). To the left of this threshold, the HR was 1.145 (95% CI 1.091–1.2, $p < 0.001$), indicating a positive association; conversely, to the right of the inflection point, the HR exhibited a non-significant decline to 1.0057 (95% CI 0.9923–1.0192, $p = 0.4064$). Notably, the association on the right side of the inflection point was not statistically significant, suggesting that for NLR values below approximately 16.475, every 1

**Table 2. Relationship between neutrophil-to-lymphocyte ratio and 90-mortality in different models.**

| Variable | Model 1 | | Model 2 | | Model 3 | | Model 4 | |
|---|---|---|---|---|---|---|---|---|
| | HR (95%CI) | *P* value | HR (95%CI) | *P* value | HR (95%CI) | *P* value | HR (95%CI) | *P* value |
| NLR count per 10 | 1.14 (1.1–1.18) | <0.001 | 1.14 (1.1~1.19) | <0.001 | 1.13 (1.09~1.18) | <0.001 | 1.11 (1.06~1.16) | <0.001 |
| Q1(0.02–3.64) | 1(Ref) | | 1(Ref) | | 1(Ref) | | 1(Ref) | |
| Q2(3.65–7.65) | 2.14 (1.15–3.96) | 0.016 | 2.16 (1.17~4.01) | 0.014 | 2.15 (1.16~3.99) | 0.015 | 2.01 (1.08~3.75) | 0.029 |
| Q3(7.66–15.45) | 3.77 (2.12–6.7) | <0.001 | 3.93 (2.2~7) | <0.001 | 3.94 (2.21~7.03) | <0.001 | 3.74 (2.08~6.73) | <0.001 |
| Q4(15.46–222.92) | 7.58 (4.37–13.12) | <0.001 | 7.6 (4.39~13.18) | <0.001 | 7.38 (4.25~12.8) | <0.001 | 6.06 (3.44~10.7) | <0.001 |
| *P* for Trend | 1.93 (1.67–2.24) | <0.001 | 1.93 (1.66~2.24) | <0.001 | 1.91 (1.64~2.21) | <0.001 | 1.78 (1.53~2.08) | <0.001 |

NLR, Neutrophil-to-Lymphocyte Ratio.

Model 1: Unadjusted.

Model 2: Adjusted for age, sex.

Model 3: Adjusted for age, sex, asthma, COPD, CHD, DM and CRF.

Model 4: Adjust for age, sex, asthma, COPD, CHD, DM, CRF, smoke, alcoholism, PLT, AST, CRE, Na and PSI.

unit increase in NLR conferred a 14.5% higher in 90-day mortality. Furthermore, beyond this threshold,90-day mortality demonstrated stabilization, but still high.

### Kaplan–Meier survival curve analysis

As depicted in the Kaplan–Meier curve (Fig 3), the 90-day cumulative survival rate of the Q4 group was significantly lower than that of the other groups ($p < 0.0001$).

### Subgroup analyses

Stratified and interaction analyses were conducted to evaluate the consistency of the association between the NLR and 90-day mortality across diverse subgroups, including age, sex, asthma, COPD, CHD, DM, smoking status, and alcohol consumption. The forest plot revealed a robust and independent association between the NLR and 90-day mortality in patients with pneumonia (Fig 4). Notably, significant interaction effects were identified for age and smoking status in relation to the NLR-mortality association ($p < 0.05$).

### Diagnostic efficiency of NLR compared with other biomarkers

We further assessed the prognostic value of NLR in patients with pneumonia. ROC curves generated for NLR and established biomarkers were illustrated in S1 Fig. The AUC value for NLR was 0.714. For predicting 90-day mortality, NLR demonstrated superior performance relative to established biomarkers, including WBC, NEUT, LYM, CRP, PCT($AUC_{WBC} = 0.607$, $AUC_{NEUT} = 0.653$, $AUC_{LYM} = 0.662$, $AUC_{CRP} = 0.529$, $AUC_{PCT} = 0.564$, $p < 0.05$). Using 9.34 as the optimal cutoff value of NLR, the sensitivity was 69.78%, and the specificity was 67.70% (S2 Table).

### Sensitivity analysis

For sensitivity analysis, the NLR was dichotomised using the optimal cutoff value of 10.0, previously established via ROC curve analysis, confirming a persistent association between NLR and 90-day mortality in patients with pneumonia (HR: 3.15, 95% CI 2.29–4.32, $p < 0.001$), as detailed in S3 Table.

### Discussion

This study demonstrates a significant association between the NLR and 90-day mortality in patients with pneumonia treated with glucocorticoids. The findings demonstrate a statistically significant elevation in 90-day mortality associated

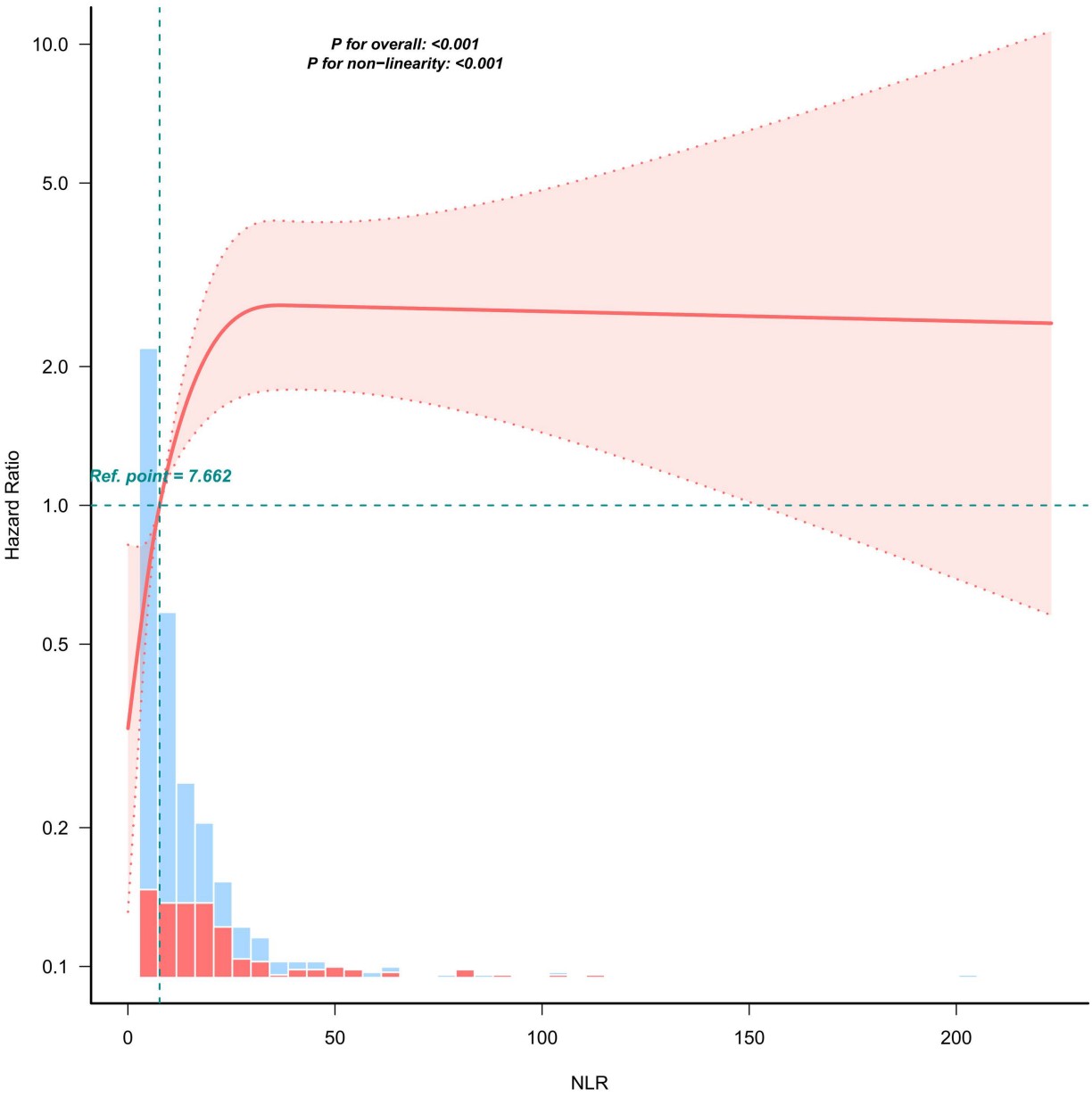

**Fig 2. Dose-response relationship between Neutrophil-to-Lymphocyte ratio and 90-day mortality.** Adjusted for age, sex, asthma, COPD, CHD, DM, CRF, smoke, alcoholism, PLT, AST, CRE, Na, and PSI.

with higher NLR values. Interestingly, the threshold effect curve analysis revealed a clear association between NLR and 90-day mortality below the inflection point. An increase of 1 unit in NLR is linked to a 14.5% increase in mortality when NLR is 16.475. Notably, there was evidence suggesting a dose-response relationship, indicating that the association between NLR and 90-day mortality exhibited a nonlinear pattern (*p* for nonlinearity <0.001). The NLR, derived from routine complete blood counts, serves as an inflammation biomarker reflecting systemic immune dysregulation. Elevated NLR values exhibit significant association with 90-day mortality in patients with pneumonia, necessitating heightened clinical attention to inflammatory status. Its utility as a rapid, cost-effective, and widely accessible biomarker is particularly

**Table 3. The non-linear relationship between Neutrophil-to-Lymphocyte ratio and 90-day mortality.**

| Threshold of Neutrophil-to-Lymphocyte ratio | HR | 95%CI | P value |
|---|---|---|---|
| <16.475 | 1.145 | 1.145 (1.091,1.2) | < 0.001 |
| ≥16.475 | 1.0057 | 1.0057 (0.9923,1.0192) | 0.4064 |
| Likelihood Ratio test | | | <0.001 |

Abbreviations: HR, hazard ratio; CI, confidence interval. HRs were adjusted for age, sex, asthma, COPD, CHD, DM, CRF, smoke, alcoholism, PLT, AST, CRE, Na and PSI. Only 99.5% of the data is displayed.

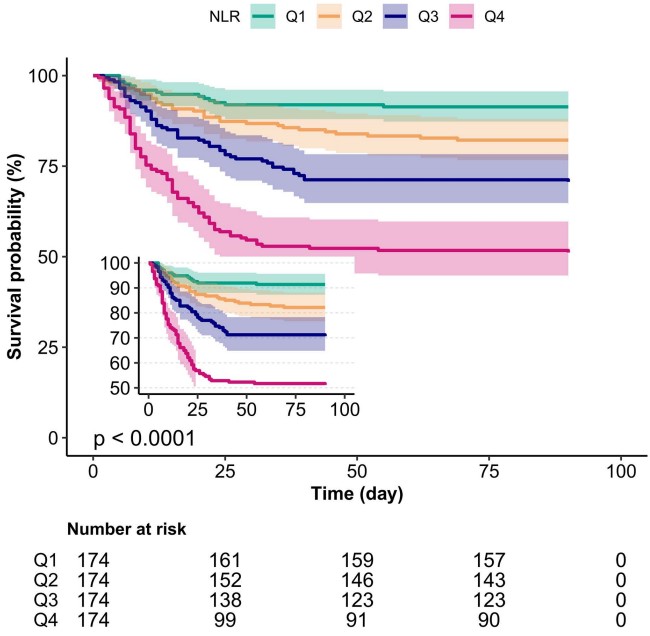

**Fig 3. Kaplan–Meier survival curves for day 90 of patients with pneumonia depending on the quartile of neutrophil-to-lymphocyte ratio. Adjusted for age, sex, asthma, COPD, CHD, DM, CRF, smoke, alcoholism, PLT, AST, CRE, Na and PSI.**

valuable in severe pneumonia, enabling prompt assessment of inflammatory burden and may provide clinicians with quick stratification. This finding suggests that elevated NLR may portend adverse hospital outcomes, potentially indicative of greater disease severity and suboptimal anti-inflammatory treatment response. Specifically, an NLR threshold ≥16.475 demonstrates significant predictive value for 90-day mortality in this population. Both the sensitivity and stratified analyses confirmed that the association between NLR and 90-day mortality remained robust. In stratified analyses, significant interaction effects were observed for age and smoking status but not for other variables. The potential interaction effects of age and smoking on the association between NLR and 90-day mortality in patients with pneumonia warrant further investigation.

The host inflammatory response is a pivotal factor in the development and progression of pneumonia. Noninvasive inflammatory biomarkers, including C-reactive protein (CRP), erythrocyte sedimentation rate (ESR), white blood cell (WBC) count, procalcitonin, interleukin (IL)-6, IL-8, interferon-alpha, and tumor necrosis factor-alpha, have been extensively utilized to enhance diagnostic accuracy. This is because the outcomes of specimen cultures as well as laboratory and radiologic assessments are not invariably informative. However, the majority of these biomarkers are costly and, as a result, are not consistently employed in routine clinical practice. Consequently, there remains an unmet need for novel

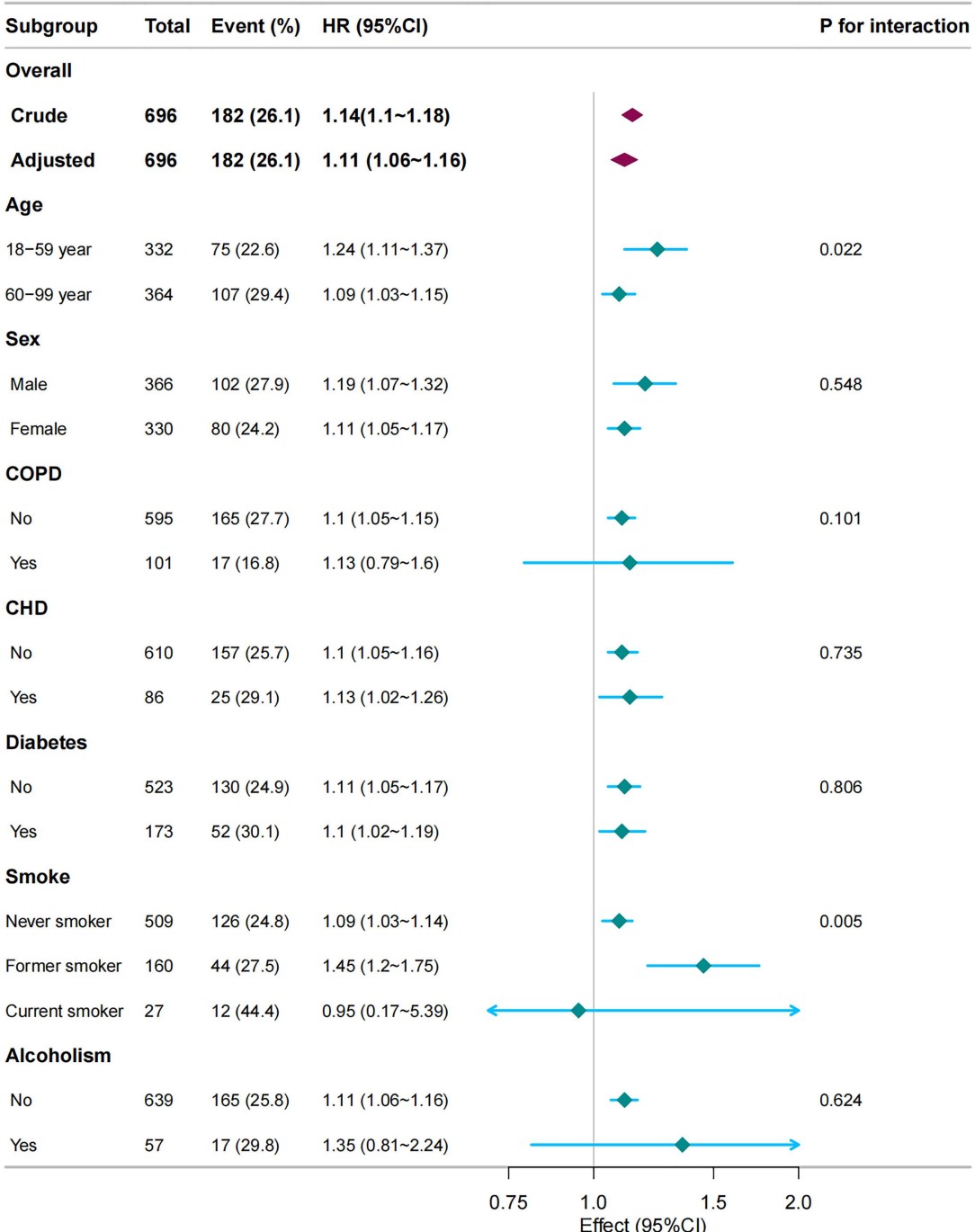

| Subgroup | Total | Event (%) | HR (95%CI) | | P for interaction |
|---|---|---|---|---|---|
| **Overall** | | | | | |
| **Crude** | **696** | **182 (26.1)** | **1.14(1.1~1.18)** | | |
| **Adjusted** | **696** | **182 (26.1)** | **1.11 (1.06~1.16)** | | |
| **Age** | | | | | |
| 18−59 year | 332 | 75 (22.6) | 1.24 (1.11~1.37) | | 0.022 |
| 60−99 year | 364 | 107 (29.4) | 1.09 (1.03~1.15) | | |
| **Sex** | | | | | |
| Male | 366 | 102 (27.9) | 1.19 (1.07~1.32) | | 0.548 |
| Female | 330 | 80 (24.2) | 1.11 (1.05~1.17) | | |
| **COPD** | | | | | |
| No | 595 | 165 (27.7) | 1.1 (1.05~1.15) | | 0.101 |
| Yes | 101 | 17 (16.8) | 1.13 (0.79~1.6) | | |
| **CHD** | | | | | |
| No | 610 | 157 (25.7) | 1.1 (1.05~1.16) | | 0.735 |
| Yes | 86 | 25 (29.1) | 1.13 (1.02~1.26) | | |
| **Diabetes** | | | | | |
| No | 523 | 130 (24.9) | 1.11 (1.05~1.17) | | 0.806 |
| Yes | 173 | 52 (30.1) | 1.1 (1.02~1.19) | | |
| **Smoke** | | | | | |
| Never smoker | 509 | 126 (24.8) | 1.09 (1.03~1.14) | | 0.005 |
| Former smoker | 160 | 44 (27.5) | 1.45 (1.2~1.75) | | |
| Current smoker | 27 | 12 (44.4) | 0.95 (0.17~5.39) | | |
| **Alcoholism** | | | | | |
| No | 639 | 165 (25.8) | 1.11 (1.06~1.16) | | 0.624 |
| Yes | 57 | 17 (29.8) | 1.35 (0.81~2.24) | | |

Effect (95%CI): 0.75   1.0   1.5   2.0

**Fig 4. Subgroup and stratified analyses of the association between Neutrophil-to-Lymphocyte ratio with 90-day mortality.** Adjusted for age, sex, asthma, COPD, CHD, DM, CRF, smoke, alcoholism, PLT, AST, CRE, Na and PSI.

biomarkers that are simple, specific, and cost-effective for the diagnosis and monitoring of pneumonia. Inflammatory biomarkers, including NLR, systemic immune-inflammation index (SII), systemic inflammation response index (SIRI), and platelet-to-lymphocyte ratio (PLR), can be used to evaluate the status of systemic inflammation and characteristics of immune response [27]. Neutrophilia and lymphocytopenia represent characteristic physiological manifestations of the innate immune system in response to systemic inflammatory processes. The pathophysiology of lymphocytopenia involves three distinct mechanisms: (1) enhanced programmed cell death of lymphocytes;(2) sequestration of these cells within the reticuloendothelial system, hepatic tissues, and visceral lymphatic networks; and (3) altered trafficking patterns throughout the lymphatic system. Neutrophilia represents the converse phenomenon during systemic inflammation, resulting from the demargination of neutrophils and the stimulation of stem cells by growth factors, specifically granulocyte-colony stimulating factor [28–30]. The NLR, which is taken in routine blood tests and can reveal the imbalance between pro-inflammatory and anti-inflammatory systems, has broad application possibilities in assessing inflammatory reactions and prognosis. It has been reported that elevated NLR serves as a reliable biomarker reflecting both disease severity and prognostic outcomes in patients with severe infections or systemic inflammatory conditions [31].

NLR has emerged as a pivotal biomarker for evaluating systemic inflammatory responses and infection susceptibility, demonstrating significant prognostic value in various clinical contexts including community-acquired pneumonia [17,32], infectious diseases [33], and intracerebral hemorrhage-related stroke outcomes [34]. Recent clinical investigations have demonstrated that the NLR exhibits superior prognostic accuracy compared with conventional infection biomarkers, including CRP, WBC count, and neutrophil count, in young adult patients presenting with community-acquired pneumonia in emergency department settings [17]. Gunay et al. pioneered the clinical application of the NLR as an inflammatory severity biomarker in patients with COPD [35], while subsequent studies further identified NLR as an independent prognostic indicator for both COPD exacerbation severity and mortality risk [36,37]. Notably, a comprehensive systematic review and meta-analysis demonstrated that NLR is significantly associated with multiple clinical outcomes in patients with acute exacerbation of COPD, encompassing clinical symptom profiles and pulmonary function parameters as well as prognostic indicators such as increased risk of bacterial infection and both short-(90-day) and long-term (24-month) mortality [38]. Xiaoyi Feng [12] demonstrated that an NLR > 14.17 at discharge was an independent risk factor for 90-day mortality in hospitalized patients with acute exacerbation of COPD, which was consistent with our study's findings, emphasizing that the NLR serves as a crucial indicator for predicting outcomes individuals diagnosed with pneumonia. The results obtained in our investigation demonstrate consistency with those reported in previous studies. Since the association between NLR and mortality in pneumonia patients has been extensively studied [17,18,32], we compared the predictive value of NLR with currently biomarkers in predicting 90-day mortality in pneumonia patients. We found that NLR performed better than WBC, NEUT, LYM, CRP and PCT.

In addition to these widely acknowledged findings, several noteworthy observations have emerged from our study. First, we consistently identified an inverted L-shaped association between NLR and 90-day mortality in patients with pneumonia taking glucocorticoids, even after adjusting for potential confounders. Consequently, elevated NLR may serve as a valuable predictor of the 90-day mortality in this population. Importantly, the magnitudes of the significant associations between NLR and incident 90-day mortality in patients with pneumonia taking glucocorticoids were also consistent across subgroups stratified by age, sex, asthma, COPD, CHD, DM, smoking, and alcohol consumption; age and smoking status had significant interaction effects($p < 0.05$).

Our study has several strengths and presents two significant methodological contributions to the field of pneumonia research. Primarily, it represents a pioneering investigation into the prognostic significance of the NLR in patients with pneumonia. We also employed advanced smooth curve fitting techniques to elucidate potential nonlinear associations between NLR levels and clinical outcomes, thereby providing a more nuanced understanding of this relationship. Furthermore, to meticulously control for potential confounding variables, we employed a robust Cox regression analysis. This approach entailed the integration of multiple models to comprehensively assess the data. In addition, we conducted subgroup analyses using meticulously defined categories to further elucidate the relationships under investigation.

However, this study has several limitations that must be acknowledged. First, due to the nature of the observational study, we only examined the correlation of NLR with 90-day mortality, and could not establish the cause-effect relationship between them. Second, despite performing regression, stratified analysis, there is still a possibility of residual confounding from unmeasured or unknown factors. Third, the limited sample size in our study may have restricted the comprehensive evaluation of statistical power and the exploration of potential interrelationships among multiple variables. While these preliminary observations may provide clinically relevant insights, external validation through larger, well-designed cohort studies is essential to establish robust evidence and strengthen the reliability of our findings. Moreover, while our findings are primarily confined to hospitalized pneumonia patients undergoing glucocorticoid-based regimens, the identified inflection point may possess broader applicability to the general population of hospitalized pneumonia patients and external validation of the 16.475 threshold will be performed in subsequent studies. Finally, it is important to note that our study predominantly included participants from China, which limits the generalizability of our results and underscores the need for parallel studies in diverse populations. In the future, prospective cohort studies or subgroups based on randomized controlled trials will be conducted to strengthen causal inference, including the use of external validate cohort to confirm our findings.

## Conclusions

We have shown that the NLR is a predictor of 90-day mortality in patients with pneumonia. A non-linear relationship was identified, indicating that high levels of NLR are associated with an increased 90-day mortality in this population, with a threshold value of 16.475. The findings presented herein warrant attention, as this association may be crucial for clinicians when considering treatment strategies aimed at reducing 90-day mortality in patients with pneumonia. It may provide clinicians with quick stratification of patients into different prognostic categories. Given the potential for confounding factors, further studies are warranted to corroborate these results. This study should serve as the foundation for future well-designed studies to evaluate the effect of elevated NLR on mortality and establish potential causal relationships.

## Supporting information

**S1 Fig.  ROC curves.**
(TIF)

**S1 Table.  Details of missing values.** Abbreviations: COPD, chronic obstructive pulmonary disease; CHD, coronary heart disease; DM, diabetes mellitus; CRF, coronary renal failure; PLT, platelet; AST, Aspartate Aminotransferase; CRE, creatinine; PSI, pneumonia severity index (PSI) score.
(TIF)

**S2 Table.  Cut-Off points of NLR, CRP, PCT, WBC, LYM and NEUT for 90-day mortality.** NLR: Neutrophil/Lymphocyte Ratio; CRP: C-Reactive Protein; PCT: Procalcitonin; WBC: White Blood Cell Count; LYM: Lymphocyte Count; NEUT: Neutrophil Count; NPV: negative predictive value; PPV: positive predictive value; AUC: area-under-curve.
(DOCX)

**S3 Table.  Sensitivity analysis.** NLR, Neutrophil-to-Lymphocyte Ratio. Model 1: unadjusted. Model 2: adjusted for age, sex. Model 3: adjusted for age, sex, asthma, COPD, CHD, DM and CRF. Model 4 adjust for age, sex, asthma, COPD, CHD, DM, CRF, smoke, alcoholism, PLT, AST, CRE, Na and PSI.
(DOCX)

## Author contributions

**Conceptualization:** Caizhen Chen.

**Data curation:** Baoli Li.

**Formal analysis:** Baoli Li, Qian Geng.

**Methodology:** Qian Geng.

**Validation:** Xiuguo Zhang.

**Writing – original draft:** Caizhen Chen, Baoli Li, Qian Geng.

**Writing – review & editing:** Xiuguo Zhang.

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
