## [Decision Letter · Decision Letter 0]

14 May 2025

Dear Dr. Chen,

Thank you for submitting your manuscript to PLOS ONE. After careful consideration, we feel that it has merit but does not fully meet PLOS ONE’s publication criteria as it currently stands. Therefore, we invite you to submit a revised version of the manuscript that addresses the points raised during the review process.

We look forward to receiving your revised manuscript.

Kind regards,

Vipula Rasanga Bataduwaarachchi, MD

Academic Editor

PLOS ONE

2. Please amend either the abstract on the online submission form (via Edit Submission) or the abstract in the manuscript so that they are identical.

3. Please remove your figures from within your manuscript file, leaving only the individual TIFF/EPS image files, uploaded separately. These will be automatically included in the reviewers’ PDF.

Additional Editor Comments:

Please attend to all questions and comments from the reviewers.

Reviewers' comments:

Reviewer's Responses to Questions

**Comments to the Author**

1. Is the manuscript technically sound, and do the data support the conclusions?

Reviewer #1: Partly

Reviewer #2: Yes

2. Has the statistical analysis been performed appropriately and rigorously?

Reviewer #1: Yes

Reviewer #2: Yes

3. Have the authors made all data underlying the findings in their manuscript fully available?

Reviewer #1: Yes

Reviewer #2: Yes

4. Is the manuscript presented in an intelligible fashion and written in standard English?

Reviewer #1: Yes

Reviewer #2: Yes

Reviewer #1: This manuscript addresses an important clinical topic: the prognostic role of the neutrophil-to-lymphocyte ratio (NLR) in predicting 90-day mortality in patients with pneumonia receiving glucocorticoids alone or in combination with other immunosuppressants. The study is well-structured and follows STROBE guidelines, and the use of advanced statistical modeling, including two-piecewise linear regression and smooth curve fitting, adds value to the investigation. The identification of a non-linear relationship with an inflection point at NLR = 16.475 is particularly noteworthy.

However, several aspects of the manuscript require clarification or improvement to enhance its scientific rigor and clinical relevance:

---

Major Comments:

1. Justification and Clinical Implication of the Inflection Point (NLR = 16.475):

While the inflection point is statistically supported, more clinical interpretation is needed. How can clinicians utilize this cutoff in practice? Consider discussing its applicability in clinical decision-making.

2. Handling of Confounding and Causal Inference:

As this is a retrospective observational study, the authors should more explicitly acknowledge and discuss the potential for residual confounding and the limitations in inferring causality, despite adjustment for multiple covariates.

3. Ethical Considerations and Informed Consent:

The manuscript mentions both anonymized datasets and informed consent. Clarify whether all patients from the Dryad database gave consent, and under what conditions. Transparency is essential in ethics reporting.

4. Subgroup and Sensitivity Analyses:

Subgroup results suggest effect modification by age and smoking. However, sensitivity analyses could be strengthened by examining other clinically relevant subgroups or using alternative cutoffs for NLR.

5. Missing Data Handling:

The authors applied multivariate imputation using a Bayesian Ridge estimator. Please detail the percentage of missing data per variable and justify why this imputation method was selected over others.

---

Minor Comments:

1. Language and Grammar:

The manuscript contains several grammatical errors and awkward phrasing. Consider professional language editing to improve readability and clarity.

2. Terminology Consistency:

Ensure consistent use of terms, especially for "neutrophil-to-lymphocyte ratio (NLR)", which is sometimes written with and without hyphens or inconsistent abbreviations.

3. Figure Legends and Interpretation:

Legends should clearly indicate all adjustments made in regression or Kaplan-Meier curves. Currently, figures lack self-contained explanations.

4. Abstract Refinement:

The abstract should explicitly state that the mortality risk plateaued beyond the NLR threshold, to aid understanding for readers without statistical expertise.

---

Conclusion:

The manuscript provides valuable insights and has the potential to contribute meaningfully to the field. However, the above revisions are necessary to improve its scientific and clinical clarity. I recommend Major Revision at this stage.

Reviewer #2: 1. Originality

While NLR has been studied in many contexts, the novelty here is slightly limited unless compared head-to-head with other biomarkers in the same population.

2. Materials and methods

A. Limited details on how missing data was handled, although Bayesian Ridge imputation is mentioned.

B. No validation cohort was used, which limits generalizability.

3. Statistically analysis

A. NLR was treated as both a continuous and categorical variable, but external validation of the threshold (16.475) is missing.

B. No use of ROC curves or AUC to assess predictive performance, which would be useful for clinical implementation.

4. Results

A. Results may not be generalizable beyond hospitalized Chinese populations or other types of immunosuppressive treatments.

B. Over interpretation risk: NLR may correlate with illness severity but isn’t necessarily causative.

5. Recommendations

• Use a prospective cohort or RCT-based subgroup for stronger causal inference.

• Include external validation cohort to confirm findings.

• Compare NLR’s prognostic value against CRP, procalcitonin, and other inflammatory markers.

• Investigate clinical cutoffs with ROC analysis to enhance practical utility.

**Do you want your identity to be public for this peer review?** For information about this choice, including consent withdrawal, please see our Privacy Policy

Reviewer #1: **Yes: ** Dr. Naif Taleb Ali

Reviewer #2: **Yes: ** "I confirm that I have no competing interests that may influence this review, received no assistance in its preparation, and affirm that this review is my own work—not signed on behalf of another person."

---

## [Author Response · Author response to Decision Letter 1]

16 Jul 2025

Revision Rebuttal

Dear Editor and reviewer:

On behalf of my co-authors, I greatly appreciate the careful review and comments from both you and the reviewers. We believe that by implementing the suggested changes, our manuscript entitled “Non-linear association between neutrophil-to-lymphocyte ratio and 90-day mortality in patients with pneumonia receiving glucocorticoids alone or in combination with other immunosuppressants: a retrospective cohort study”(Number: PONE-D-25-10696) has improved and is more suitable for submission to“PLOS ONE”.

Herein, we have included point-by-point responses for each of the comments in the attached document and have revised our manuscript accordingly. The revised sections were highlighted in YELLOW.

Furthermore, the manuscript has been proofread native English-speaking editor from a professional English language editing company, thus improving the overall language and reliability of the manuscript.

We hope this revised manuscript has addressed your concerns and look forward to your positive responses to the revised manuscript.

The authors declare no conflict of interest regarding this work. All authors have read the revised manuscript and approved it for submission to “PLOS ONE”. Please do not hesitate to contact us if we can be of any further assistance.

Thank you and best regards.

The main corrections in manuscript and the responses to the reviewer's comments are as followings:

Reviewer #1: 

comment 1

1. Justification and Clinical Implication of the Inflection Point (NLR = 16.475):While the inflection point is statistically supported, more clinical interpretation is needed. How can clinicians utilize this cutoff in practice? Consider discussing its applicability in clinical decision-making.

Response:

Thank you for your professional review recommendations;As you mentioned,how the inflection point value of NLR is applied in clinical practice is indeed of crucial importance.Based on your review comments, we have conducted an in-depth discussion on the significance of NLR inflection point values in clinical practice, the revised content of is as follows:“The NLR may provide clinicians with quick stratification and disease severity assessment of patients into different prognostic categories.By evaluating the risk of 90-day mortality depending on NLR,when NLR is less than 16.475,patients should receive airway management and early anti-infection intervention to avoid deterioration,especially targeted antimicrobial treatment should be started as early as possible.When NLR is higher than 16.475,required more need for mechanical ventilation assistance,as well as a greater proportion of systemic administration of antibiotics,or even admission to a respiratory intensive care unit (RICU),and multidrug-resistant pathogens must be considered when selecting antimicrobial agents for pneumonia in patients who are receiving high-dose steroids or in those with persistent lymphocytopenia.All in all, we should emphasize monitoring and intervention for patients with high NLR levels during hospitalization.Both the sensitivity and stratified analyses confirmed that the association between NLR and 90-day mortality remained robust.In stratified analyses, significant interaction effects were observed for age and smoking status but not for other variables.”

We have amended this in the manuscript and have highlighted it in yellow.Please see line 226-238 page 11-12 in the revised manuscript.please review again.

comment 2

2. Handling of Confounding and Causal Inference:

As this is a retrospective observational study, the authors should more explicitly acknowledge and discuss the potential for residual confounding and the limitations in inferring causality, despite adjustment for multiple covariates.

Response:

Thank you for your positive advice.We have amended this in the manuscript and have highlighted it in yellow.Please see line 300-303 page 14 in the revised manuscript.

comment 3

3. Ethical Considerations and Informed Consent:

The manuscript mentions both anonymized datasets and informed consent. Clarify whether all patients from the Dryad database gave consent, and under what conditions. Transparency is essential in ethics reporting.

Response:

Thank you for your valuable suggestion.We have updated the Ethics Statement to clarify that the original study (Li et al. 2020) received ethical approval from China-Japan Friendship Hospital (Approval No. 2015-86) with a waiver of informed consent, as it used only anonymized retrospective data. Our secondary analysis adhered to the same conditions, using fully anonymized data without identifiable information, thus requiring no additional consent. This is now explicitly stated in the Materials and Methods section (page 4,lines 74).

Exclusion criterion (2) was revised to improve precision.Please see page 5,line 96.

comment 4

4. Subgroup and Sensitivity Analyses:

Subgroup results suggest effect modification by age and smoking. However, sensitivity analyses could be strengthened by examining other clinically relevant subgroups or using alternative cutoffs for NLR.

Response:

We are grateful for your insightful suggestions.According to your advice,we conducted a sensitivity analysis using alternative NLR cut off value 10.0, validating the outcome data,Specifically as follows:

(1) Statistical analysis section:“To evaluate the robustness of our findings, we performed a sensitivity analysis by reclassifying NLR using the optimal cutoff value of 10.0, as established in prior studies[17,32].”(page 6-7,lines 141-143).

(2) Result section:“Sensitivity analysis

For sensitivity analysis, the NLR was dichotomised using the optimal cutoff value of 10.0, previously established via ROC curve analysis,confirming a persistent association between NLR and 90-day mortality in patients with pneumonia (HR: 3.15, 95% CI 2.29-4.32, P<0.001), as detailed in S3 Table.(page11,lines 215-218).

(3)Discussion section:“Both the sensitivity and stratified analyses confirmed that the association between NLR and 90-day mortality remained robust”(page12,lines 235-236).

S2 Table Sensitivity analysis

Variable Model 1 Model 2 Model 3 Model 4

HR (95%CI) P value HR (95%CI) P value HR (95%CI) P value HR (95%CI) P value

NLR

Q1(<10) 1(Ref) 1(Ref) 1(Ref) 1(Ref)

Q2(≥10) 3.5 (2.58~4.74) <0.001 3.56 (2.62~4.84) <0.001 3.54 (2.6~4.83) <0.001 3.15 (2.29~4.32) <0.001

NLR,Neutrophil-to-Lymphocyte Ratio.

Model 1:unadjusted.

Model 2:adjusted for age,sex.

Model 3:adjusted for age,sex,asthma,COPD,CHD,DM and CRF.

Model 4 adjust for age, sex, asthma,COPD,CHD,DM,CRF, smoke,alcoholism,PLT,AST,CRE,Na and PSI.

comment 5

5. Missing Data Handling:

The authors applied multivariate imputation using a Bayesian Ridge estimator. Please detail the percentage of missing data per variable and justify why this imputation method was selected over others.

Response:

Thank you for your comments.

We have supplemented the list of missing values. The details of the missing value are shown in Supplementary S1 Table .Supplements were made in the revised manuscript. The results are as follows:

S1 Table Details of missing values.

Variables The number of missing values The percent of

missing values (%)

  Age 0 0

  Sex, n (%) 0 0

  Asthma, n (%) 0 0

  COPD, n (%) 0 0

  CHD, n (%) 0 0

  DM, n (%) 0 0

  CRF, n (%) 0 0

  Smoke, n (%) 0 0

  Alcoholism, n (%) 0 0

  PLT×109/L 11 1.5805

  AST(U/L) 11 1.5805

  CRE(μmmol/L) 9 1.2931

  Na(mmol/L) 14 2.0115

  PSI 0 0

Abbreviations:COPD,chronic obstructive pulmonary disease;CHD,coronary heart disease; DM, diabetes mellitus;CRF,coronary renal failure;PLT,platelet;AST,Aspartate Aminotransferase; CRE, creatinine;PSI,pneumonia severity index (PSI) score

Our study conducted multivariate imputation using a Bayesian Ridge estimator.Its main virtues are that imputations are restricted to the observed values and that it can preserve non-linear relations even if the structural part of the imputation model is wrong. Finally, the function mice.impute.sample just takes a random draw from the observed data, and imputes these into missing cells. This function does not condition on any other variable.The idea of MI is to take into account uncertainty in predicting missing values by creating multiple complete datasets.

Multiple imputation (MI) is an advanced method in handling missing values. In contrast to single imputation, MI creates a number of datasets (denoted by m) by imputing missing values. That is, one missing value in original dataset is replaced by m plausible imputed values. These values take imputation uncertainty into consideration.

Minor Comments:

Comment 1

1. Language and Grammar:

The manuscript contains several grammatical errors and awkward phrasing. Consider professional language editing to improve readability and clarity.

Response:

Thank you for your positive advice.We have corrected the grammar mistakes and inappropriate expressions.The English content in the paper has been professionally edited and checked by native English speakers.Please review again.

Comment 2

2. Terminology Consistency:

Ensure consistent use of terms, especially for "neutrophil-to-lymphocyte ratio (NLR)", which is sometimes written with and without hyphens or inconsistent abbreviations.

Response:

Thank you for your valuable suggestion.In accordance with your recommendation,we have thoroughly rechecked the the consistent use of terms,especially for "neutrophil-to-lymphocyte ratio (NLR)".And we have made the necessary corrections to ensure consistency throughout the manuscript.Please review again.

Comment 3

3. Figure Legends and Interpretation:

Legends should clearly indicate all adjustments made in regression or Kaplan-Meier curves. Currently, figures lack self-contained explanations.

Response:

Thank you for your valuable suggestion.Following your advice,I have supplemented the legend of the Kaplan-Meier curve.The revised content is as follows:“Adjusted for age, sex, asthma, COPD, CHD, DM, CRF, smoke,alcoholism,PLT,AST,CRE,Na and PSI.”Specifically,it is on page 19,line 455,and highlighted in yellow.

Comment 4

4. Abstract Refinement:

The abstract should explicitly state that the mortality risk plateaued beyond the NLR threshold, to aid understanding for readers without statistical expertise.

Response:

Thank you for your valuable suggestion to improve the quality of our manuscript.Based on your proposal,we have added it in the Abstract of revised manuscript as“after exceeding the threshold, the risk of 90-day mortality tended to stabilize”,Specifically,it is on page 2,line 35-36,and highlighted in yellow.

Reviewer #2: 

Comment 1

1. Originality

While NLR has been studied in many contexts, the novelty here is slightly limited unless compared head-to-head with other biomarkers in the same population.

Response:

Thank you for your constructive suggestion.According to your suggestion,We further explored the prognostic value of NLR in pneumonia patients.Receiver operator characteristic (ROC) curves constructed for NLR and other existing biomarkers are presented in S1 Fig 1.

Modifications and supplements have been made to the Abstract, Statistical Methods, Results, and Discussion sections of the manuscript.

Specifically:

①Abstract: The supplementary contents are as follows:“Furthermore, the ROC curve analysis revealed that NLR was the optimal biomarker for predicting 90-day mortality with an AUC of 0.714 (95% CI:0.670-0.757).Using 9.34 as the cutoff value of NLR, the sensitivity was 69.8%, and the specificity was 67.7%.”(page 2, lines 30-32).

②Statistical Methods:The supplementary contents are as follows:“The Youden index was utilized to calculate the cut-off value of each variable,and receive operating characteristics (ROC) curve analysis was used to assess the sensitivity and specificity of different biomarkers of adverse prognosis.The areas under the curve (AUC) was used to evaluate prognostic accuracy,with comparisons performed using a nonparametric approach.Higher AUC values indicate greater discriminatory ability”(page 6 and lines 137-141).

③Results: The supplementary contents are as follows:“Diagnostic efficiency of NLR compared with other biomarkers.

We further explored the prognostic value of NLR in pneumonia patients.Receiver operator characteristic (ROC) curves constructed for NLR and other existing biomarkers are presented in S1 Fig.The areas under the curve(AUC) values were 0.714 for NLR. Compared to existing biomarkers, such as WBC, NEUT,LYM,CRP,PCT,NLR performed better in predicting 90-day mortality(AUCNLR=0.714,AUCWBC=0.607,AUCNEUT=0.653,AUCLYM=0.662,AUCCRP=0.529,AUCPCT=0.564,p<0.05).Using 9.34 as the optimal cutoff value of NLR, the sensitivity was 69.78%, and the specificity was 67.70%(S2 Table ).”(page 10-11, lines 208-214).

④Discussion: The supplementary contents are as follows:“Since the association between NLR and mortality in pneumonia patients has been extensively studied[17-18,32],we compared the predictive value of NLR with currently biomarkers in predicting 90-day mortality in pneumonia patients.We found that NLR performed better than WBC,NEUT,LYM,CRP and PCT.(page 13-14, lines 279-282).

Comment 2

2. Materials and methods

A. Limited details on how missing data was handled, although Bayesian Ridge imputation is mentioned.

B. No validation cohort was used, which limits generalizability.

Response :

A.Thank you for your valuable advice.We have taken your feedback into account and have revised the description in the manuscript.The revised contents are as follows:“We used multiple imputations, based on 5 replications and a chained equation approach method in the R MI procedure, to maximize statistical power and minimize bias that might occur to account for missing data.The details of the missing value are shown in S1 Table.”(page 7,line 143-146).

B.Thank you for your constructive suggestion.External validation is explained in the limitations of the article, see page 15, lines 312-314, which is also a direction for our future research.

Comment 3

3. Statistically analysis

A. NLR was treated as both a continuous and categorical variable, but external validation of the threshold (16.475) is missing.

B. No use of ROC curves or AUC to assess predictive performance, which would be useful for clinical implementation.

Response :

A.Thank you for your comment.We appreciate your professional advice,external validation of the inflection point value of 16.475 will be a direction of our research and we will take this as our next task.

B.To test the predictive performance, we added the ROC curve and the area under the curve,as detailed S1 Fig.

Modifications and supplements have been made to the Abstract, Statistical Methods, Results, and Discussion sections of the manuscript. Specifically:

①Abstract: The supplementary contents are as follows:“Furthermore, the ROC curve analysis revealed that NLR was the optimal biomarker for predicting 90-day mortality with an AUC of 0.714 (95% CI:0.670-0.757).Using 9.34 as the cutoff value of NLR, the sensitivity was 69.8%, and the specificity was 67.7%.”( page 2, lines 30-32).

②Statistical Methods: The supplementary contents are as follows:“The Youden index was utilized to calculate the cut-off value of each variable,and receive operating characteristics (ROC) curve analysis was used to assess the sensitivity and specificity of different biomarkers of adverse prognosis.The areas under the curve (AUC) was used to evaluate prognostic accuracy,with comparisons performed using a nonparametric approach.Higher AUC values indicate greater discriminatory ability.”(page 6,lines 137-141).

③Results: The supplementary contents are as follows:“Diagnostic efficiency of NLR compared with other biomarkers

We further explored the prognostic value of NLR in pneumonia patients.Receiver operator characteristic (ROC) curves constructed for NLR and other existing biomarkers are presented in S1 Fig.The areas under the curve(AUC) values were 0.714 for NLR. Compared to existing biomarkers, such as WBC, NEUT,LYM,CRP,PCT,NLR performed better in predicting 90-day mortality(AUCNLR=0.714, AUCWBC= 0.607, AUCNEUT=0.653, AUCLYM=0.662, AUCCRP=0.529, AUCPCT=0.564, p<0.05).Using 9.34 as the optimal cutoff value of N

---

## [Editor Report · Decision Letter 1]

21 Jul 2025

Non-linear association between neutrophil-to-lymphocyte ratio and 90-day mortality in patients with pneumonia receiving glucocorticoids alone or in combination with other immunosuppressants: a retrospective cohort study

PONE-D-25-10696R1

Dear Dr. Chen,

We’re pleased to inform you that your manuscript has been judged scientifically suitable for publication and will be formally accepted for publication once it meets all outstanding technical requirements.

Kind regards,

Vipula Rasanga Bataduwaarachchi, MD

Academic Editor

PLOS ONE

---

## [Editor Report · Acceptance letter]

PONE-D-25-10696R1

PLOS ONE

Dear Dr. Chen,

I'm pleased to inform you that your manuscript has been deemed suitable for publication in PLOS ONE. Congratulations! Your manuscript is now being handed over to our production team.

Kind regards,

on behalf of

Dr. Vipula Rasanga Bataduwaarachchi

Academic Editor

PLOS ONE